# THE VENDISCOPE: AN ALGORITHMIC MICROSCOPE FOR DATA COLLECTIONS

## ABSTRACT

The evolution of microscopy, beginning with its invention in the late 16th century, has continuously enhanced our ability to explore and understand the microscopic world, enabling increasingly detailed observations of structures and phenomena. In parallel, the rise of data-driven science has underscored the need for sophisticated methods to explore and understand the composition of complex data collections. This paper introduces the *Vendiscope*, the first *algorithmic microscope* designed to extend traditional microscopy to computational analysis. The Vendiscope leverages the Vendi scores – a family of differentiable diversity metrics —- and assigns weights to data points based on their contribution to the overall diversity of the collection. These weights enable high-resolution data analysis at scale. We demonstrate this across biology and machine learning (ML). We analyzed the 250 million protein sequences in the protein universe, discovering that over 200 million are near-duplicates and that ML models like AlphaFold fail on proteins with Gene Ontology (GO) functions that contribute most to diversity. Additionally, the Vendiscope can be used to study phenomena such as memorization in generative models. We used the Vendiscope to identify memorized training samples from 13 different generative models spanning several model classes and found that the best-performing generative models often memorize the training samples that contribute least to diversity. Our findings demonstrate that the Vendiscope can serve as a powerful tool for data-driven science, providing a systematic and scalable way to identify duplicates and outliers, as well as pinpointing samples prone to memorization and those that models may struggle to predict—even before training.

## 1 INTRODUCTION

As machine learning (ML) continues to become more deeply integrated in critical applications, the ability to scrutinize models and the data they are trained on becomes more important (Biderman et al., 2024; Alampara et al., 2025; Banerjee et al., 2024; Longpre et al., 2024). Current evaluation practices, however, are dominated by performance benchmarking. While convenient for comparison, these metrics do not enable deeper analysis into the contents of a dataset or the failure patterns of a model. For example, protein structure prediction models have achieved remarkable progress on the CASP leaderboards (Kryshtafovych et al., 2019; 2021), but solely monitoring predictive accuracy will not reveal where models like AlphaFold systematically struggle.

To address this gap, this paper introduces the concept of *algorithmic microscopes*, tools designed to reveal hidden structure in both dataset composition and model behavior. An algorithmic microscope emphasizes understanding – helping researchers understand the contents of their data and where their models fail. Given the breadth of ML applications, such a tool must be flexible across domains. To this end, we present the Vendiscope, a scalable algorithmic microscope for analyzing models and datasets in any domain where similarity can be defined. The Vendiscope uses the probability-weighted Vendi Score (pVS) (Friedman & Dieng, 2023) to measure the contribution of each datapoint to the overall diversity of a collection. This is done by assigning each data point with an unknown weight, using those weights to define the pVS of the set of data points, and maximizing the pVS to learn the weights. Those weights in turn are used to analyze data and model outputs.

**Contributions.** We make several contributions in this paper as detailed below.

- We demonstrate the Vendiscope's contribution scores can help identify outliers and near-duplicates in linear time.
- We show how the same framework can be used to evaluate machine learning models – both predictive and generative. For predictive models, the Vendiscope can correlate performance metrics with contribution to diversity, thus helping characterize data points where models perform poorly. For generative models, the Vendiscope is the first method to fully characterize the types of data points that are prone to memorization as those that contribute least to the diversity of the training data.
- We apply the Vendiscope to the 250 million sequences composing the protein universe, where it identifies >80% redundant data points at a 90% similarity threshold. It also uncovers AlphaFold's failure in modeling sequences that contribute most to the diversity of the protein universe. When applied to 13 generative models trained on CIFAR-10, the Vendiscope uncovers a consistent relationship between rarity and memorization, revealing that models achieving the highest perceptual quality do so by duplicating and memorizing samples that contribute least to diversity.

## 2 THE VENDISCOPE

We first provide background on the pVS as a measure of diversity. Next, we present the Vendiscope's optimization algorithm for measuring datapoint-level contributions to diversity, followed by an analysis of its complexity and implementation. Finally, we describe how the Vendiscope's outputs can be interpreted, which will allow us to evaluate datasets and models in Section 3.

### 2.1 PROBABILITY-WEIGHTED VENDI SCORES

Consider a collection of $N$ elements $(\boldsymbol{x}_1, \ldots, \boldsymbol{x}_N)$. Let $k(\cdot, \cdot)$ denote a positive semi-definite kernel that measures the similarity between any two elements, and such that $k(\boldsymbol{x}_i, \boldsymbol{x}_i) = 1 \ \forall i$. Denote by $\boldsymbol{K}$ the similarity matrix induced by the kernel $k(\cdot, \cdot)$. Its element at row $i$ and column $j$ is $K_{ij} = k(\boldsymbol{x}_i, \boldsymbol{x}_j)$. Since $k(\cdot, \cdot)$ is positive semi-definite, $\boldsymbol{K}$ is positive semi-definite and has non-negative eigenvalues which we denote by $\lambda_1, \ldots, \lambda_N$. Let $\boldsymbol{p} = (p_1, \ldots, p_N)$ denote a discrete probability distribution over the collection $(\boldsymbol{x}_1, \ldots, \boldsymbol{x}_N)$. Define $\tilde{\boldsymbol{K}}_p = \operatorname{diag}(\sqrt{\boldsymbol{p}}) \boldsymbol{K} \operatorname{diag}(\sqrt{\boldsymbol{p}})$ and let $\eta_{1p}, \ldots, \eta_{Np}$ denote the eigenvalues of $\tilde{\boldsymbol{K}}_p$. Friedman & Dieng (2023) define the pVS of the collection as the exponential of the Shannon entropy of the eigenvalues. This can be generalized using the Rényi entropy (Pasarkar & Dieng, 2024),

$$\mathrm{pVS}_k(\boldsymbol{x}_1, \ldots, \boldsymbol{x}_N, \boldsymbol{p}) = \exp\left(\frac{1}{1-q} \log \sum_{i \in \mathrm{supp}(\eta)} \eta_{ip}^q\right), \tag{1}$$

where $\mathrm{supp}(\eta)$ denotes the set of non-zero eigenvalues of $\tilde{\boldsymbol{K}}_p$ and $q \geq 0$ is the order of the pVS.

### 2.2 MEASURING DIVERSITY CONTRIBUTIONS

We can use the pVS as an objective function to measure the contribution of each datapoint to the dataset's overall diversity. In particular, the Vendiscope considers $\boldsymbol{p}$ as an unknown probability distribution to be learned by maximizing Eq. 1,

$$\boldsymbol{p}^* = \arg\max_{\boldsymbol{p}} \ \mathrm{pVS}_k(\boldsymbol{x}_1, \ldots, \boldsymbol{x}_N, \boldsymbol{p}) \text{ such that } \sum_{i=1}^{N} p_i = 1. \tag{2}$$

Optimizing over the pVS balances the spectrum of the probability-weighted similarity matrix. This amplifies dimensions of the dataset that would be otherwise underrepresented. As a result, the solution to Eq. 2 will lead to higher probabilities on the rarest samples, and lower probabilities on the most common ones (Section A.1).

The Vendiscope's gradient-based algorithm is provided in Algorithm 1. Following the computation of the VSs and its gradients, we perform projected gradient decent using the active set method described in Michelot (1986).

---

**Algorithm 1** The Vendiscope: An algorithmic microscope for data collections

---

Inputs: Data $\{\boldsymbol{x}_1, \ldots, \boldsymbol{x}_n\}$, similarity kernel $k$, order $q > 0$, step sizes $\epsilon_1, \ldots, \epsilon_n$

Form a data matrix $\boldsymbol{X} \in \mathbb{R}^{n \times d}$, normalize its rows: $\boldsymbol{X}_i = \boldsymbol{x}_i / ||\boldsymbol{x}_i||_2$, and initialize diversity contribution scores uniformly $p_i = \frac{1}{n}$ for all $i = 1, \ldots, n$

**while** *not converged* **do**

    Compute weighted similarity matrix $\tilde{K} = \begin{cases} \boldsymbol{X}^\top \mathrm{diag}(\sqrt{p}) \mathrm{diag}(\sqrt{p}) \boldsymbol{X} & \text{if } k \text{ is cosine} \\ \mathrm{diag}(\sqrt{p}) \boldsymbol{K} \mathrm{diag}(\sqrt{p}) & \text{otherwise} \end{cases}$

    Compute loss function $\mathcal{L}(p) = -\log \mathrm{pVS}_k(\boldsymbol{x}_1, \ldots, \boldsymbol{x}_n)$

    Compute gradients $\nabla_{p_1} \mathcal{L}(p), \ldots, \nabla_{p_n} \mathcal{L}(p)$ using backpropagation

    Compute unnormalized weights $y_1, \ldots, y_n$ such that $y_i = p_i - \epsilon_i \nabla_{p_i} \mathcal{L}(p)$

    Set $v_i = y_i$ for all $i$ and $\rho = \frac{1}{n} \sum_{i=1}^n y_i - 1$

    **while** *the norm of v continues to change* **do**

        Set $v_i = \mathbb{I}(y_i > \rho)$ and $\rho = \frac{\sum_{i=1}^n v_i - 1}{\sum_{j=1}^n v_j}$ for all $i \in \{1, \ldots, n\}$

    **end**

    Update diversity contribution scores $p_i = \max(y_i - \rho, 0)$ for all $i \in \{1, \ldots, n\}$

**end**

---

**Time and space complexity.** Each iteration of the Vendiscope requires calculating the VS for a collection of $n$ elements, which involves computing the eigenvalues of an $n \times n$ matrix. This process has a time complexity of $O(n^3)$. However, Friedman & Dieng (2023) indicated that when data embeddings are available, the VS can be computed cheaply by using a cosine similarity kernel with corresponding similarity matrix $\boldsymbol{K} = \boldsymbol{X}^T \boldsymbol{X}$, where $\boldsymbol{X} \in \mathbb{R}^{n \times d}$ denotes the data embedding matrix. In this case the VS computation has complexity $O(d^2 n + d^3)$. The improvement in complexity enables the scaling of the Vendiscope to large collections where $n \gg d$.

The projected gradient updates are linear in $n$ as well. Condat (2016) notes that the active set method has an observed runtime of $O(n)$. There are certain examples for which the runtime can be quadratic (Cominetti et al., 2014), but we will not encounter such instances when most weights are similar. Empirically, small learning rates ensure linear runtimes. In all, we reach a time complexity of $O(d^3 + d^2 n + n) = O(d^2 n)$ and a space complexity of $O(dn)$ for each iteration of the Vendiscope.

## 2.3 IMPLEMENTATION DETAILS

The Vendiscope enables the scalable analysis of large data collections. Below we describe the design choices that drive its effectiveness.

**Scaling to massive datasets.** As presented, Algorithm 1 would require the entire dataset to be loaded into memory. This is prohibitively expensive for many of the massive datasets available today. We circumvent this problem by estimating the pVS using only a subset of the data's dimensions at each iteration. In particular, at each iteration $t$, we sample a random subset of the columns of the data matrix, $d_t \subseteq \{1, \ldots, D\}$ and use $X_{d_t} \in \mathbb{R}^{n \times |d_t|}$ instead of the entire dataset $X$. This approach provides an approximation of the true pVS. Our subsampling approach also allows us to take advantage of data parallelism by sampling a separate set of data dimensions for each GPU. These approaches allow us to run the Vendiscope on datasets with hundreds of millions of samples.

**Hyperparameters and convergence.** The Vendiscope requires the Vendi score order $q$ as a user-specified hyperparameter. Previous work by Pasarkar et al. (2023) demonstrated that small values $q < 1$, are more sensitive to rare elements, whereas large values of $q$ place greater emphasis on common elements. We find that the sensitivity of small values $q$ helps all elements have non-zero contributions to diversity. Figure 6 illustrates this behavior in a synthetic 2D example. For finite values of $q$, the Vendiscope remains sensitive to individual samples, with this effect strongest for small $q < 1$. Only in the limit $q = \infty$ does the method behave differently, as the Vendiscope depends solely on the largest eigenvalue and effectively ignores the dataset's smaller modes. We therefore use $q = 0.1$ and $q = 0.5$ in all experiments.

---

**Algorithm 2** Efficient near-duplicate detection with the Vendiscope

---

Inputs: Data $\{x_1, \ldots, x_n\}$ sorted in order of the Vendiscope scores, similarity kernel $k$, near-duplicate similarity threshold $s \leq 1$, and search-range $m \leq n$

**for** $i = 1, \ldots, n$ **do**
  If $x_i$ is in a cluster $c \in C$ then $x_i$ is already analyzed, go to the next sample
  Else create new cluster $c \leftarrow \{\mathbf{x}_i\}$
  **for** $j = i + 1, \ldots, i + m$ **do**
    If $k(x_i, x_j) > s$ and $x_j$ not in a cluster then $c = c \cup x_j$
  **end**
  Add $c$ to a list of clusters $C$
**end**

---

The choice of kernel function is also important, as the Vendiscope's notion of rarity is determined by the patterns of similarity induced by the kernel. Figures 7 and 8 show this on ImageNet classes. We compare the Vendiscope's results when using a cosine kernel with embeddings from the DINOv2 (Oquab et al., 2023) and Inception-V3 (Szegedy et al., 2016) network, as well as when using color-based kernels. We show that different feature representations and similarity functions can provide distinct rarity rankings.

In the presented analyses, we focus on the relative ranking of elements based on the Vendiscope weights rather than the magnitude of the weights themselves. As a result, we stop the Vendiscope when the ranking of elements stabilizes. This occurs within 500 iterations in all of our studies.

**Initialization and identifiability.** In Algorithm 1, we initialize the weights to be equal, reflecting, in a Bayesian sense, an uninformative prior over the Vendiscope's probabilities. This choice allows the Vendiscope to assign identical weights to exact duplicates. A random weight initialization would cause issues with the identifiability of exact duplicates due to the optimization of the pVS. For exact duplicates, an optimized pVS only places a constraint on the sum of their scores. Consider, for instance, a collection with 3 elements and the kernel matrix with the first column $(1, 0, 0)$ and and identical second and third columns $(0, 1, 1)$. In this setting, the pVS can be maximized with $p_1 = 0.5$ and $p_2 + p_3 = 0.5$, yielding an optimal pVS of 2. If we initialize all weights to be equal, $p_2$ and $p_3$ will have identical gradients throughout the iterations and will remain equal.

## 2.4 UTILIZING THE VENDISCOPE SCORES

The Vendiscope scores enable a range of diagnostic tasks. Below we highlight three uses illustrative use cases, showing how the scores act as an algorithmic microscope for datasets and models.

**Detecting rare elements.** We call *rare* elements those data points that contribute most to the diversity of the collection. As demonstrated earlier, these are the data points to which the Vendiscope assigns the highest probabilities. Our experiments also show that these data points tend to be the ones that models may struggle to predict. Instead, we find that data points that are assigned the lowest probabilities by the Vendiscope yield the best model predictions.

**Detecting duplicates.** Duplicates in data will contribute to the diversity of a dataset almost identically. These duplicates should therefore have very similar probabilities. This insight motivates how we detect duplicates in Algorithm 2. Importantly, we do not need to calculate all $N^2$ pair-wise similarities in the data and can instead focus on data points that the Vendiscope assigns similar probabilities. More specifically, we find redundant data points by only computing similarities between each sample and its $m$ closest neighbors, where closeness is measured using the assigned probabilities from the Vendiscope. Choosing $m$ large comes at a higher computational cost. We find that values of $m$ in the order of $1 - 2\%$ of the size of the dataset are sufficient for analyzing large-scale datasets with hundreds of millions of data points. At this scale, the Vendiscope can identify over $95\%$ of all duplicates at a fraction of the cost of computing all pairwise similarities.

This algorithm is also amenable to computing optimizations. After computing the Vendiscope weights, we can distribute subsets of the dataset across independent processes, avoiding the need to load the full dataset into memory. Batch comparisons can further leverage GPU matrix operations

for additional speedups. Together, these optimizations make duplicate detection with the Vendiscope both scalable and memory-efficient, suitable for large modern ML datasets.

**Detecting memorization.** Detecting whether a generative model has memorized its training data is typically done by comparing each generated output against all training examples. This brute-force strategy is prohibitively expensive for large-scale datasets. The Vendiscope offers a scalable alternative: by applying it to the training data, we find that outputs assigned the lowest probabilities are exactly those that overlap most with the generated set. We confirm this empirically in Section 3, where we show that low-probability training datapoints coincide with memorized samples across multiple image generative models. The patterns also hold when applying the Vendiscope to the generated collection instead of the training set. The generated outputs assigned the lowest probabilities by the Vendiscope have higher similarities with samples in the training set. These findings can allow researchers to more efficiently detect memorization in generative models.

## 3 EXPERIMENTS

We demonstrate the various capabilities by analyzing the protein universe and AlphaFold's performance. We then analyze CIFAR-10 and 13 image generative models trained on it. In all settings, the Vendiscope uncovers important insights about training data composition and the performance of models trained on these datasets. We also analyze benchmark materials science data alongside 3 property prediction models in Section A.3.

### 3.1 EXPLORING THE LANDSCAPE OF THE PROTEIN UNIVERSE

The UniProt database is the community's most comprehensive representation of the protein universe, containing over 250 million annotated sequences. It underpins nearly all modern ML models for proteins, including AlphaFold, ProtBert, and ProtT5 (Jumper et al., 2021; Brandes et al., 2022; Elnaggar et al., 2021) and has become an important resource for biological discovery.

We use the Vendiscope to analyze UniProt, revealing key insights into rare and redundant sequences in the dataset and how it can affect model performance. Using ProtT5 embeddings, the Vendiscope analyzes the entire database in under two hours on a single compute node equipped with 8 NVIDIA A6000 GPUs. We expect even faster speeds with optimized data loading procedures. All experimental settings are in Section A.2.

**The Vendiscope scores measure more than prevalence.** Here we show how the ranking produced by the Vendiscope can reflect important factors about how datasets are formed. In the protein universe in particular, the Vendiscope's scores capture evolutionary phenomena. We demonstrate this with two contrasting sets of proteins in Figure 1.

Proteins involved in amino acid metabolism are consistently ranked low by the Vendiscope. These proteins come from enzymes that have been repeatedly reused across different biological functions (Jensen, 1976). As a result, many homologous sequences with high similarity exist, even if they are functionally distinct. The presence of these similar sequences makes these proteins common from the perspective of the Vendiscope, driving their scores down. Binding proteins such as chemokine or bombesin receptor ligands, in contrast, are marked as rare by the Vendiscope. Each of these proteins tends to be highly distinct from each other and subject to strong evolutionary constraints that prevent the emergence of close variants with different functions (Wang et al., 2016; Zlotnik et al., 2006). The lack of similar sequences makes these protein rare, and thus they contribute strongly to the diversity of the protein universe. We provide additional examples of how common proteins are often associated with fundamental metabolic pathways in Fig. 9.

**AlphaFold struggles with proteins that contribute most to diversity.** This distinction between common and rare proteins has direct implications for model evaluation. We find that the rare sequences identified by the Vendiscope – the sequence that contribute most to the diversity of the protein universe – are also those on which AlphaFold performs most poorly. As shown in Figure 2, prediction confidence, as determined by the average predicted local distance difference test (pLDDT) over each sequence, declines significantly for the rarest sequences. Structural accuracy is

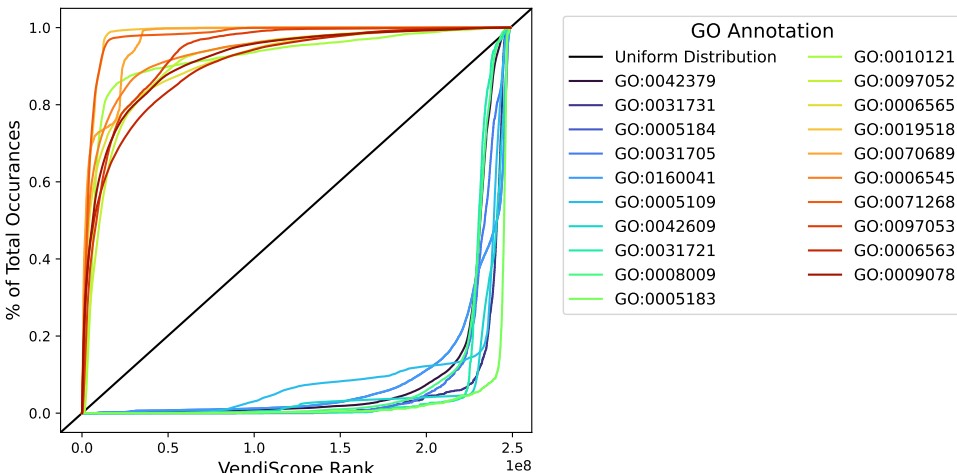

Figure 1: Various selected Gene Ontology (GO) functions that are enriched among highly-ranked and low-ranked proteins. All displayed functions concentrated in rare proteins have roles in protein binding (GO:0005515), whereas all displayed functions in low-ranked proteins fall under amino acid metabolic processes (GO:0006520).

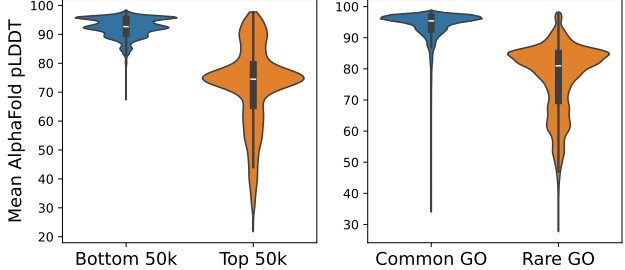

Figure 2: AlphaFold confidence is significantly worse on rare protein sequences. Left: Violin plot of average pLDDT for the top (most rare) and bottom (most common) 50,000 sequences. Right: Violin plot of AlphaFold confidences for proteins with certain GO functions. We select 10 GO functions that are primarily present among low-scoring proteins ('Common GO') and 10 GO functions that are enriched among high-scoring proteins ('Rare GO'). GO functions are shown in Fig. 1.

particularly poor for functions concentrated in rare sequences, such as binding proteins, compared to those found among common sequences. These results underscore the value of algorithmic microscopy: the same scoring that highlights outliers in the data also pinpoints where models are most likely to fail. Our analysis also provides a roadmap for improved data collection that prioritizes regions of the protein universe where new data would most enhance model performance.

**The Vendiscope efficiently detects redundant protein sequences.** Next, we deploy the Vendiscope to identify near-duplicate sequences in the protein universe. Detecting and removing redundant samples is important for building smaller versions of datasets like UniProt. For biologists, this can enable faster sequence searches and more efficient model training (Sieber et al., 2018; Suzek et al., 2015). Currently, MMseqs2 is the most popular approach for protein sequence clustering (Steinegger & Söding, 2018). In Figures 3 and 10, we highlight how the duplicate clusters identified by the Vendiscope have clear biological interpretations and are significantly larger than those identified by MMseqs2. Indeed, we identify $21,003,854$ clusters containing $210,372,272$ proteins with the Vendiscope, while MMseqs2 identifies $29,540,400$ clusters that encompass only $127,545,233$ proteins. We further benchmark the quality of clusters using GO annotations and find that the Vendiscope and MMseqs2 provide similar levels of consistency (Section A.2). The Vendiscope also runs in

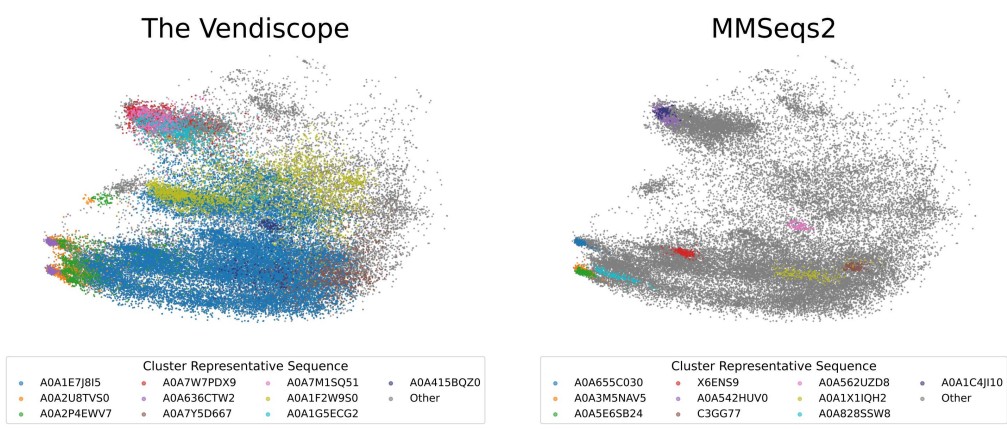

Figure 3: The Vendiscope identifies large protein clusters with consistent annotations. Top: PCA scatter plot of all proteins originating from the ahcY gene, with duplicate clusters from the Vendiscope (left) and MMseqs2 (right) overlaid. The 10 clusters with the most proteins from the ahcY gene are shown for both methods.

the same time as MMseqs2 (two hours on 40 CPU cores). By producing larger, biologically meaningful clusters without additional cost, the Vendiscope offers a scalable and practical alternative for redundancy detection in massive protein datasets.

## 3.2 DIAGNOSING IMAGE GENERATIVE MODELS

We apply the Vendiscope to CIFAR-10 and to the outputs of 13 state-of-the-art generative models trained on CIFAR-10. These models span architectures, including GANs, diffusion, and flow networks (Stein et al., 2023). In this setting, the Vendiscope exposes near-duplicates in both training and generated data and reveals systematic memorization patterns in high-performing models. All experimental settings are in Section A.4.

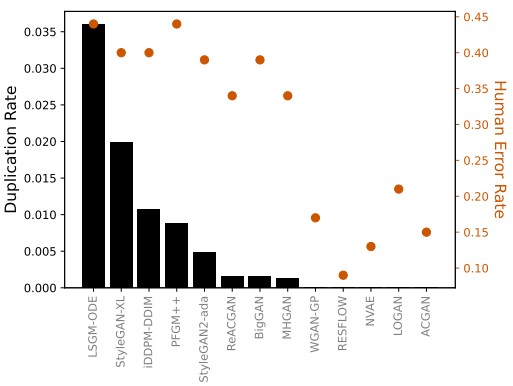

Figure 4: CIFAR-10 image generative models with high duplication rates have high human error rates. Models that produce 0 duplicates produce lower quality outputs according to human judges.

**Detecting duplicates in CIFAR-10.** The Vendiscope efficiently identifies near-duplicates in CIFAR-10 (Fig. 13), assigning them nearly identical contributions to dataset diversity. While duplicates can be identified with brute-force searches or manual curation (Recht et al., 2018), the Vendiscope provides a scalable alternative for much larger datasets.

**Detecting duplicates in state-of-the-art image generative models.** We next apply the Vendiscope to the generative models from Stein et al. (2023). *Useful* generative models should produce images that are novel, diverse, and of high perceptual quality. However, existing evaluation methods for image generative models do not directly measure the number of duplicates in the generated outputs. In Fig. 4, we show the number of duplicates for each model, as well as the average human error rate provided by Stein et al. (2023). We observe that the generative models producing the highest quality images, with lower human error rates, also produce many duplicates. These results are consistent with Pasarkar & Dieng (2024), where they found that the models that produced the highest perceptual quality models also seemed to generate large clusters of similar images.

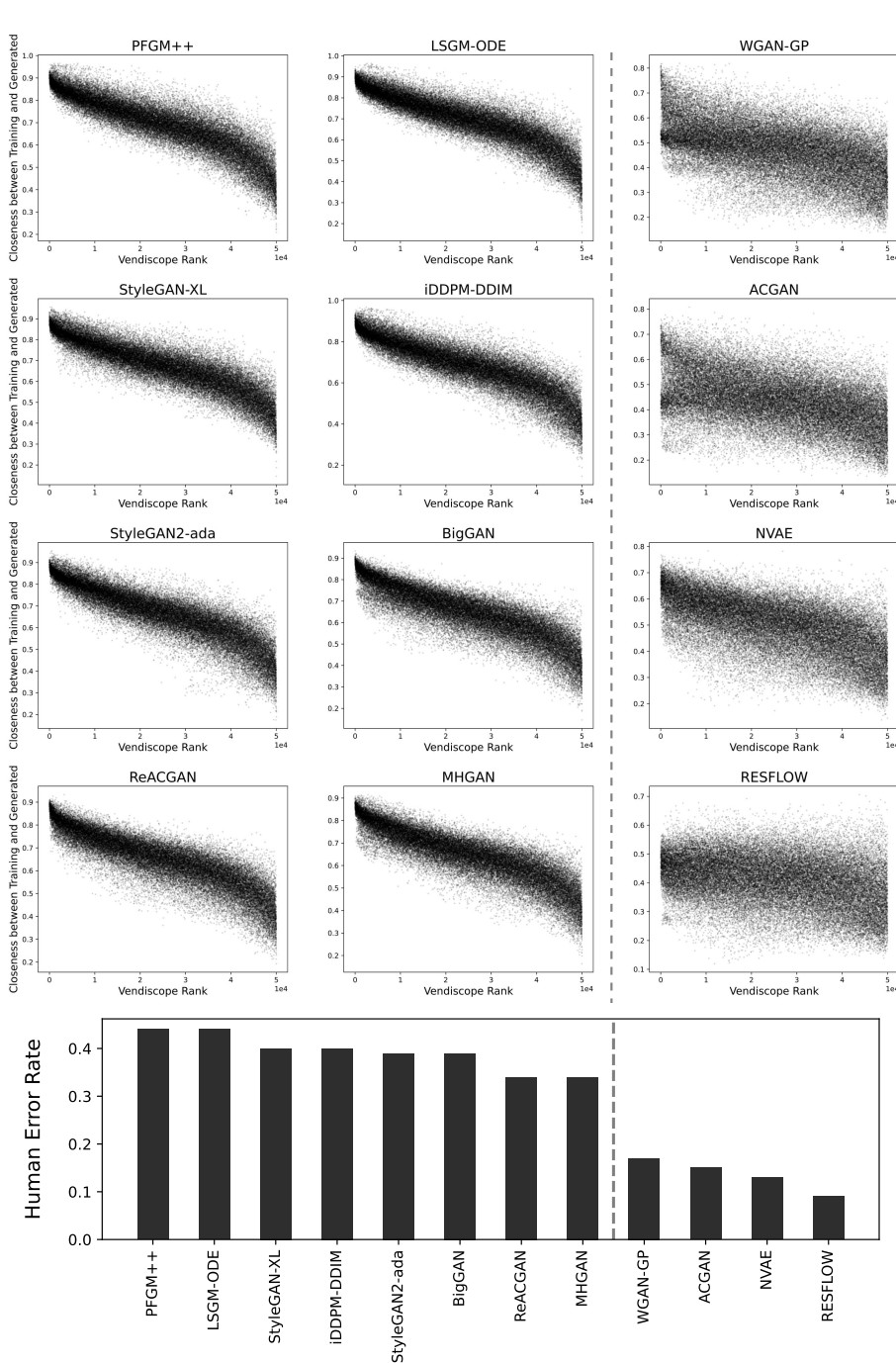

Figure 5: Memorization is strongly correlated with the Vendiscope rank of CIFAR-10 training data for various image generative models. Top: 12 models and their degree of memorization for each training point is displayed, showing strong correlations between rarity and memorization. Models for which the correlation is weaker are in the third column. Bottom: Bar plot showing that the models with stronger memorization and higher correlation with the Vendiscope rank produce higher quality images than others.

**Detecting memorization.** We now apply the Vendiscope to detect memorization on these models. Memorization is an undesirable property of generative models, although its causes are not well understood. We use the Vendiscope to study this phenomenon by comparing each training datapoint's score with its maximum similarity to generated outputs. Memorized points are those with near-duplicate matches in the generated collection. Across the 13 generative models, we find a strong negative correlation between the Vendiscope scores and memorization: low-scoring training samples are frequently reproduced by the models, while high-scoring (rare) samples are never memorized (Fig. 5, Fig. 14). This suggests that generative models preferentially copy data points that contribute least to diversity. We additionally find that models trained on CIFAR-10 which memorize common images the most achieve higher image quality scores as measured by human error rate (Figs. 5 and 15). This raises significant concerns about the reliability of image fidelity metrics like human error rate and underscores the need for more granular analyses when evaluating models.

## 4 RELATED WORK

The Vendiscope offers many capabilities, including detection of duplicates and memorized samples. Related works address these tasks individually and in specific domains. We review them below.

**Near duplication detection.** Several methods exist to detect duplicates in specific domains, e.g. proteins and text (Kocetkov et al., 2022; Lee et al., 2021; Steinegger & Söding, 2018; Zhang et al., 2023). We provide a summary of many popular algorithms in Table 1. For proteins, MMSeqs2 relies on k-mer matching (Steinegger & Söding, 2018), but its heuristics can miss near-duplicates (Ou et al., 2023). knnProtT5 leverages embeddings instead to perform k-nearest neighbors, though it struggles with variable cluster sizes (Schütze et al., 2022). For text, MinHash-based LSH (Lee et al., 2021; Kocetkov et al., 2022) scales well but ignores semantic similarity. RETSim accounts for semantic similarity by training specialized text encoders, but is not generally applicable (Zhang et al., 2023). In contrast, the Vendiscope identifies near-duplicates efficiently across domains and provides insights into datasets beyond redundancy.

**Detecting memorization in generative models.** Significant efforts have been made to identify the causes of memorization in generative models (Kandpal et al., 2022; Lee et al., 2021; Tirumala et al., 2022). Duplication and overfitting are often linked to memorization, although models may still memorize in the absence of duplicates or long training regimes (Jagielski et al., 2022; Somepalli et al., 2023). Webster et al. (2021) showed that when face datasets contain over-represented identities, generative models often reproduce those identities, revealing how redundant regions of the training distribution are prone to memorization. Our results align with this view: we find that redundant samples, those that contribute least to diversity, are more prone to memorization.

**Characterizing large-scale datasets.** Datasets like the Stack, FineWeb, and C4, have become staples for training large language models (Kocetkov et al., 2022; Penedo et al., 2024; Raffel et al., 2020). However, the contents of these datasets are not well understood. Prior work has focused on high-level analyses, such as ablations to justify curation strategies (Penedo et al., 2024), $n$-gram and duplicate counts (Elazar et al., 2023), or topic distributions (Zhong et al., 2024). The Vendiscope can complement these analyses by providing information about sample rarity. Furthermore, the Vendiscope can facilitate more nuanced duplicate searches and is applicable across domains.

**Vendi scoring.** The Vendiscope maximizes the pVS (Friedman & Dieng, 2023) and, as such, relates to methods that leverage the VS. Berns et al. (2023) optimized the sum of the pVS and the Shannon entropy of the probabilities involved in the computation of the pVS to balance the modes of generative models, enhancing their ability to produce diverse outputs. The VS has been extended and applied in multiple ways, owing to its flexibility (Askari Hemmat et al., 2024; Kannen et al., 2024; Liu et al., 2024; Nguyen & Dieng, 2024; Mousavi & Khalili, 2024; Pasarkar et al., 2023; Rezaei & Dieng, 2025; Bhardwaj et al., 2025; Jung et al., 2025). The Vendiscope optimizes the pVS via projected gradient descent, yielding interpretable sample-level measurements and an ability to scale to massive datasets with parallelization.

## 5 CONCLUSION

We introduced the Vendiscope, an algorithmic microscope designed to enhance our ability to analyze complex datasets and models. The Vendiscope measures the contribution of each datapoint to the overall diversity of the dataset in linear time. Our experiments on proteins, images, and materials show this flexible framework can identify rare and redundant data, diagnose model failure modes, and detect memorization. Looking ahead, the Vendiscope has the potential to serve as a predictive tool that helps researchers anticipate model performance even before training begins.

## CODE AND DATA AVAILABILITY

All code, data, and model checkpoints are available at this anonymized Google Drive folder.

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

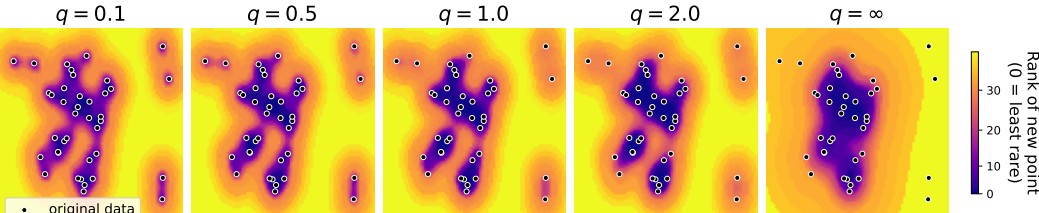

Figure 6: The Vendiscope is robust to the choice of Vendi Score order $q$ for finite values. Each panel shows a dataset of 49 points (black) together with a heatmap indicating the Vendiscope rank of a hypothetical 50th point as a function of its location. For finite $q$, the Vendiscope incorporates smaller eigenvalues, making the rarity score sensitive to the proximity to individual samples, with this effect strongest for small $q < 1$. For $q = \infty$, however, the score is determined by the largest eigenmode of the dataset, which makes rarity determined by distance from the main mode rather than on isolated datapoints.

## A  APPENDIX

### A.1  THE VENDISCOPE MEASURES CONTRIBUTION TO DIVERSITY

We argue that optimizing Eq. 2 yields probabilities that correspond to the rarity of each sample. Indeed, for all orders of $q$, Equation 2 is maximized when the eigenvalues $\eta_{1p}, \ldots, \eta_{Np}$ are the same. Therefore, to successfully optimize Equation 2 the probabilities should be learned such that all eigenvalues are within a small $\epsilon$ distance of each other: $\eta_{\min} \leq \eta_{ip} \leq \eta_{\min} + \epsilon$, where $\epsilon > 0$ and $\eta_{\min}$ denotes the minimum eigenvalue. We assume without loss of generality that $\eta_{\min}$ is non-zero. We can link the uniformity of the eigenvalues to the Vendiscope's learned probabilities using the Gershgorin Circle Theorem (Varga, 2011). From this theorem, we know that the eigenvalues of $\tilde{K}_p$ are located in discs with radii determined by the row-sums. Define $C_j = \sum_{i \neq j}^{N} K_{ij}\sqrt{p_i}$, which corresponds to a sum of weighted similarities between one sample and the rest of the dataset. Then, for each eigenvalue $\eta_{ip}$, there exists a row index $j \in \{1, \ldots, N\}$ such that

$$|\eta_{ip} - p_j| \leq \sqrt{p_j}C_j \tag{3}$$

Varga (2011) additionally states in Theorem 1.6 that if a set of $L$ discs is disjoint from all other discs, it must contain $L$ eigenvalues. As a result, if there exists a single sample $x_j$ with disc centered at $p_j$ that is disjoint from all other discs and is not within $\sqrt{p_j}C_j$ of the eigenvalue interval $[\eta_{\min}, \eta_{\min} + \epsilon]$, it would contain an eigenvalue that violates our uniformity assumption. We therefore expect all discs to be tightly clustered around the eigenvalue interval.

In order to construct such discs, the highest probabilities $p_j$ must be assigned to the samples $x_j$ with the smallest weighted row-sums $C_j$. Otherwise, any disc with small $C_j$ and $p_j$ will have a small radius and be far away from the eigenvalue interval, creating a disjoint disc. Since samples with low $C_j$ are those that are most distinct from the rest of the dataset, particularly other high-probability samples, assigning high probability to them ensures the rarest samples receive the greatest weight in the optimal $p^*$.

### A.2  PROTEIN UNIVERSE ANALYSIS

#### A.2.1  EXPERIMENTAL SETTINGS

We use the UniProtKB release v2024.02. To extract protein embeddings from sequences, we average over all per-residue embeddings from the ProtT5-XL-UniRef50 model (Elnaggar et al., 2021) to obtain a single vector respresentation per protein. We then use Vendi Score order $q = 0.1$ to calculate the Vendi Scores in 1. Finally, To detect duplicates in Algorithm 2, we use a search range of $m = 2,000,000$.

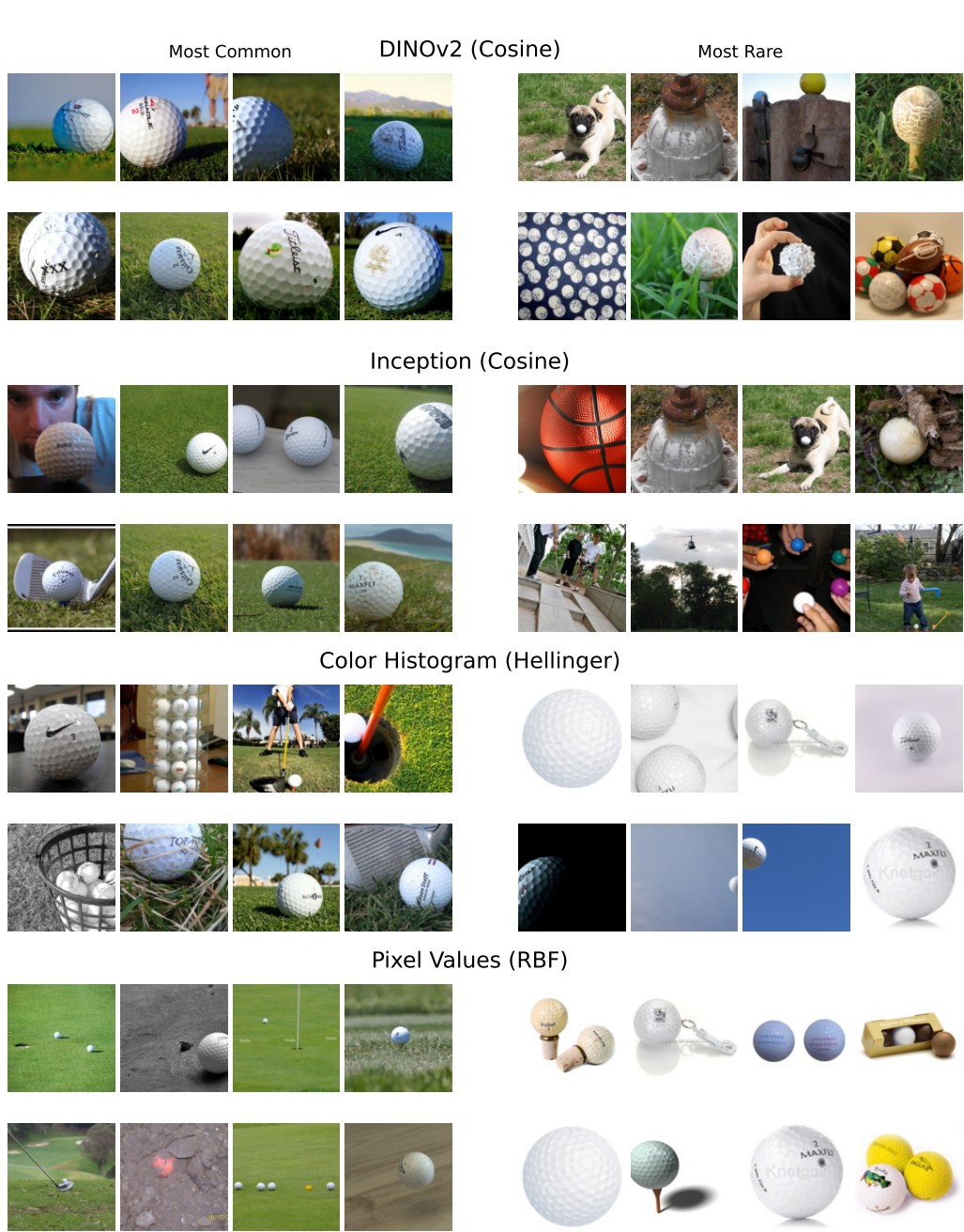

Figure 7: The Vendiscope measures rarity based on the chosen similarity function. Each panel shows the 8 most common (left) and most rare (right) images from the ImageNet `golf ball` class under different feature representations and kernel functions. When using embedding models such as DINOv2 or Inception, the Vendiscope identifies rare samples that are semantically distinct. Color-based features combined with the hellinger kernel and radial basis function (RBF) kernel focus on the background and foreground colors.

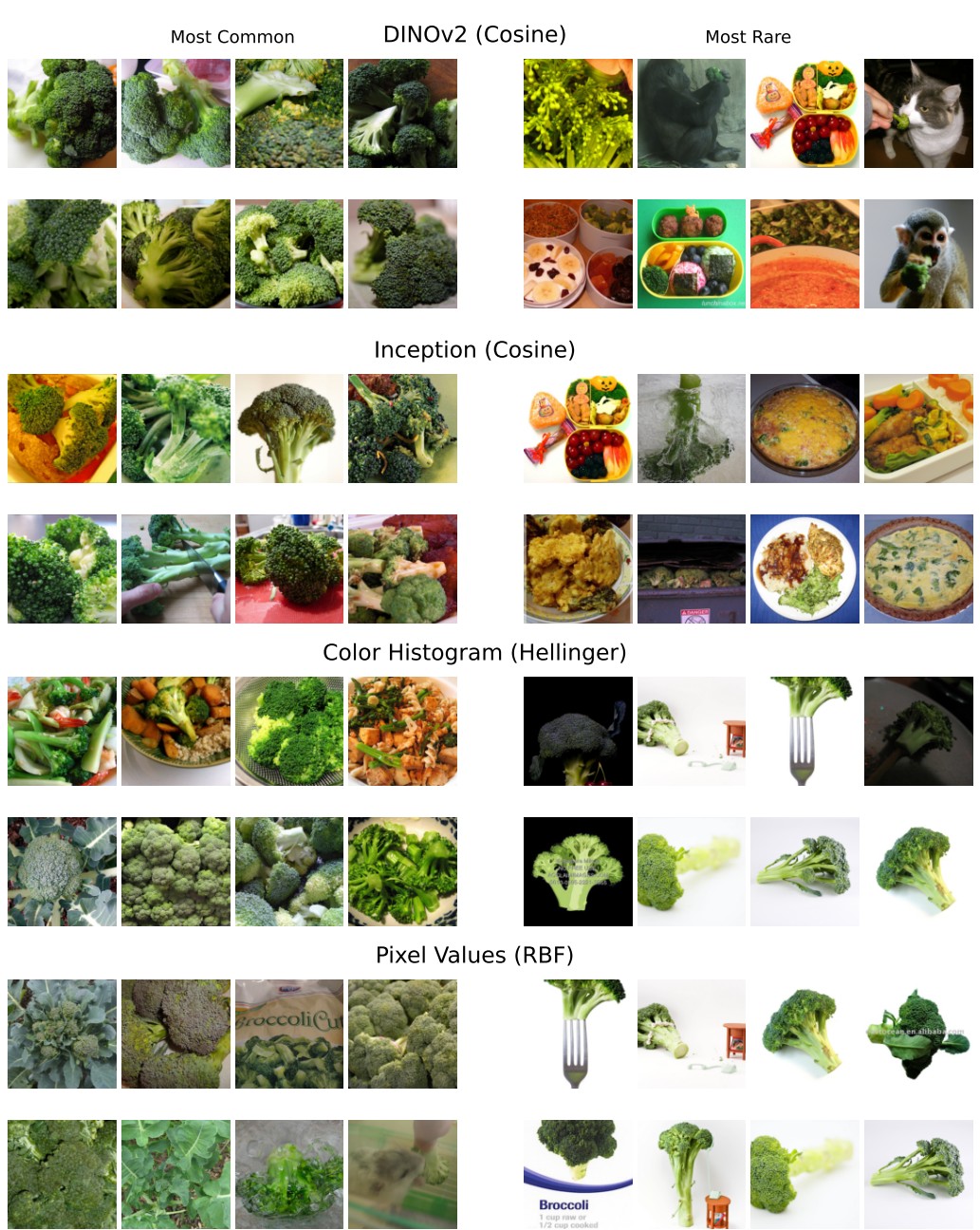

Figure 8: The Vendiscope measures rarity based on the chosen similarity function. Each panel shows the 8 most common (left) and most rare (right) images from the ImageNet `broccoli` class under different feature representations and kernel functions. When using embedding models such as DINOv2 or Inception, the Vendiscope identifies rare samples that are semantically distinct. Color-based features combined with the hellinger kernel and radial basis function (RBF) kernel focus on the background and foreground colors.

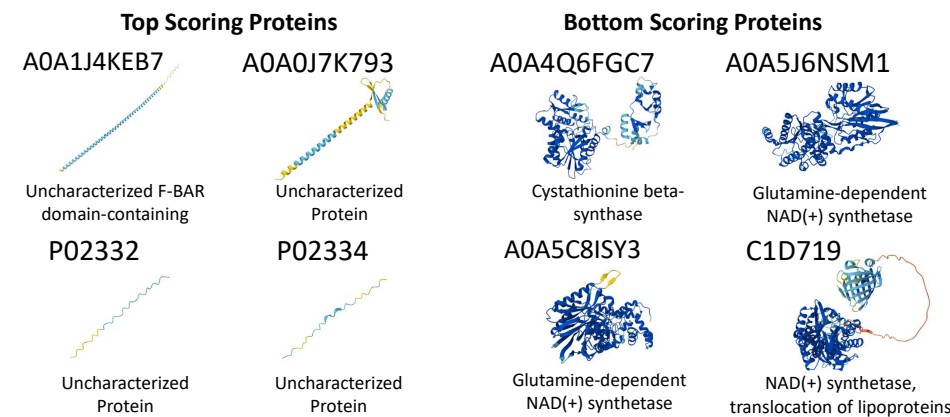

Figure 9: The Vendiscope's rarest (top scoring) proteins and those that contribute least to diversity (low scoring) proteins and their corresponding AlphaFold predicted structures. Rare proteins are mostly uncharacterized or contain unrealistic structures, such as missing the characteristic banana shape of the F-BAR domain. Bottom-scoring proteins are involved in fundamental pathways such as NAD(+) synthesis and transsulfuration.

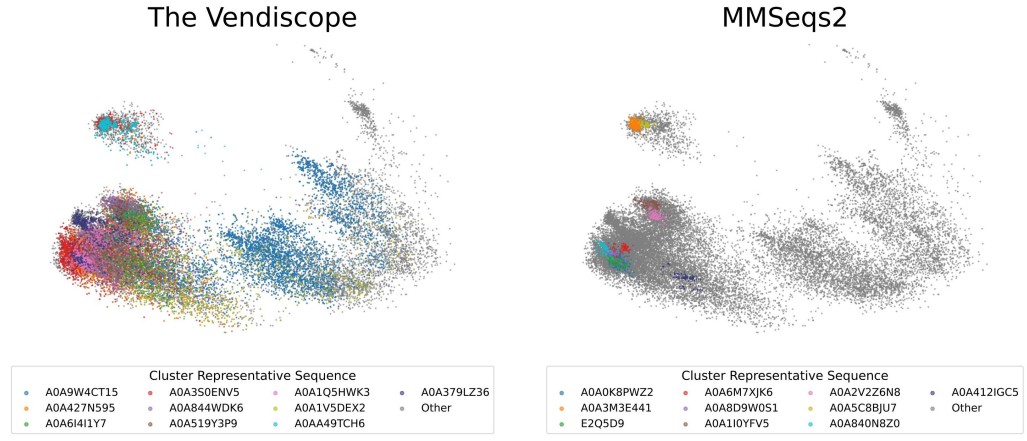

Figure 10: PCA scatter plot of all proteins originating from the ahcY gene, with duplicate clusters from the Vendiscope (left) and MMseqs2 (right) overlaid. The 10 clusters with the most proteins from the ahcY gene are shown for both methods.

### A.2.2 ADDITIONAL ANALYSIS OF THE PROTEIN UNIVERSE

We have reported that over 80% of the UniprotKB has a near-duplicate in the database based on a similarity threshold of 0.9. We find that there remains a large number of duplicates for other thresholds as well: 46.9% of sequences have a near-duplicate even for a threshold of 0.99 (Fig. 11).

To further benchmark the quality of the clusters identified by the Vendiscope, we measure the consistency of the functions of the proteins in each cluster. Each protein has a list of GO annotations that describe all of the protein's known functions (Ashburner et al., 2000; Aleksander et al., 2023). To measure the similarity between two GO terms, we record the reciprocal of the distance between GO terms on the corresponding GO tree, as described in (Sangar et al., 2007). To then compute the similarity between pairs of proteins $P_1$ and $P_2$, we must compare two lists of GO terms. We use the Average-Best-Match approach by Zhao & Wang (2018). Suppose $P_1$ has $m$ terms $\text{go}_{11}, \text{go}_{12}, \ldots, \text{go}_{1m}$ and $P_2$ has $n$ terms $\text{go}_{21}, \text{go}_{22}, \ldots, \text{go}_{2n}$. The similarity between $P_1$ and $P_2$ is defined as

$$k'(P_1, P_2) = \frac{1}{m+n} \left( \sum_{i=1}^{m} \max_{\text{go}_{1i}} k(\text{go}_{1i}, \text{go}_{2j}) + \sum_{j=1}^{n} \max_{\text{go}_{2j}} k(\text{go}_{1i}, \text{go}_{2j}) \right). \quad (4)$$

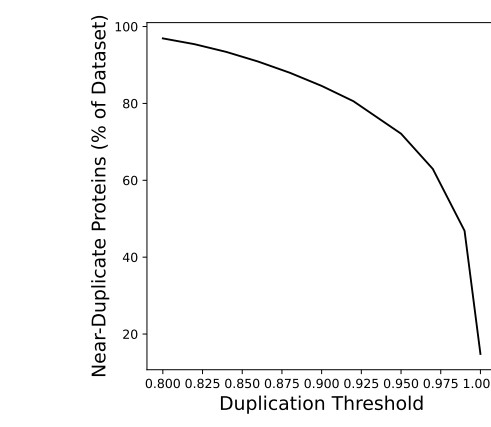

Figure 11: Number of Sequences with Near-Duplicates in the UniprotKB using different similarity thresholds. Even for larger similarity thresholds, including exact matches, there is a significant amount of near-duplication

where $k(\cdot, \cdot)$ is a similarity kernel. Eq. 4 provides a measure of similarity between the functional annotations of a pair of proteins. To then obtain a measure of consistency for a given cluster, we compute the average semantic similarity between a cluster's representative sequence and all other sequences in the cluster. In the Vendiscope, we define the representative sequence for each cluster as the sequence whose closest to the cluster's centroid. For context, the representative sequence in a MMseqs2 cluster is the cluster's longest sequence. We find that the clusters identified by the Vendiscope have an average semantic similarity of $0.942 \pm 0.105$, while those identified by MMseqs2 have an average of $0.985 \pm 0.049$ semantic similarity. Both similarities are quite high and likely suffer from how certain proteins may have poor annotations. Nevertheless, the Vendiscope is within one standard deviation of MMseqs2 in terms of semantic similarity while still identifying 65% more proteins with near-duplicates.

### A.3 Analyzing The Materials Project Database

We use the Vendiscope to analyze the composition of the Materials Project database (v2024.12.18). The Materials Project is the result of a significant computational effort to calculate the properties of many materials (Jain et al., 2013). This database has been instrumental in training ML models for materials property prediction and continues to grow. The prioritization of which materials are added has significant implications for the quality of future models. Using the Vendiscope on three popular models—ALIGNN (Choudhary & DeCost, 2021), CGCNN (Xie & Grossman, 2018), and DeeperGATGNN (Omee et al., 2022)—we characterize the materials in the Materials Project, reveal potential biases within the database, and identify patterns of model failure for property prediction.

**Material Property Prediction Model Training.** We train 3 models on the Materials Project. We use the recommended settings from each model for pre-processing crystal structures. We therefore use a cut-off radius of 8 Å for constructing graphs for CGCNN and DeeperGATGNN, and 4 Å for constructing graphs for ALIGNN. We sweep over hyperparameters such as the number of hidden layers and hidden dimensions before training models on the entire dataset. All models are trained to convergence: CGCNN uses 1000 epochs with batch-size 256, DeeperGATGNN uses a batch-size of 100 for 400 epochs, and ALIGNN uses a batch-size of 16 for 300 epochs. We use the model checkpoint at the final epoch for all downstream analysis.

**Property prediction accuracy degrades on materials that enhance diversity.** The three selected models all achieved state-of-the-art property prediction performance at the time of their publication. However, they all fail to model the same types of materials: the ones that enhance diversity.

We trained each model to predict formation energy and band gap. We then extracted embeddings from each model by using the output from the layer just before the final prediction layer and used

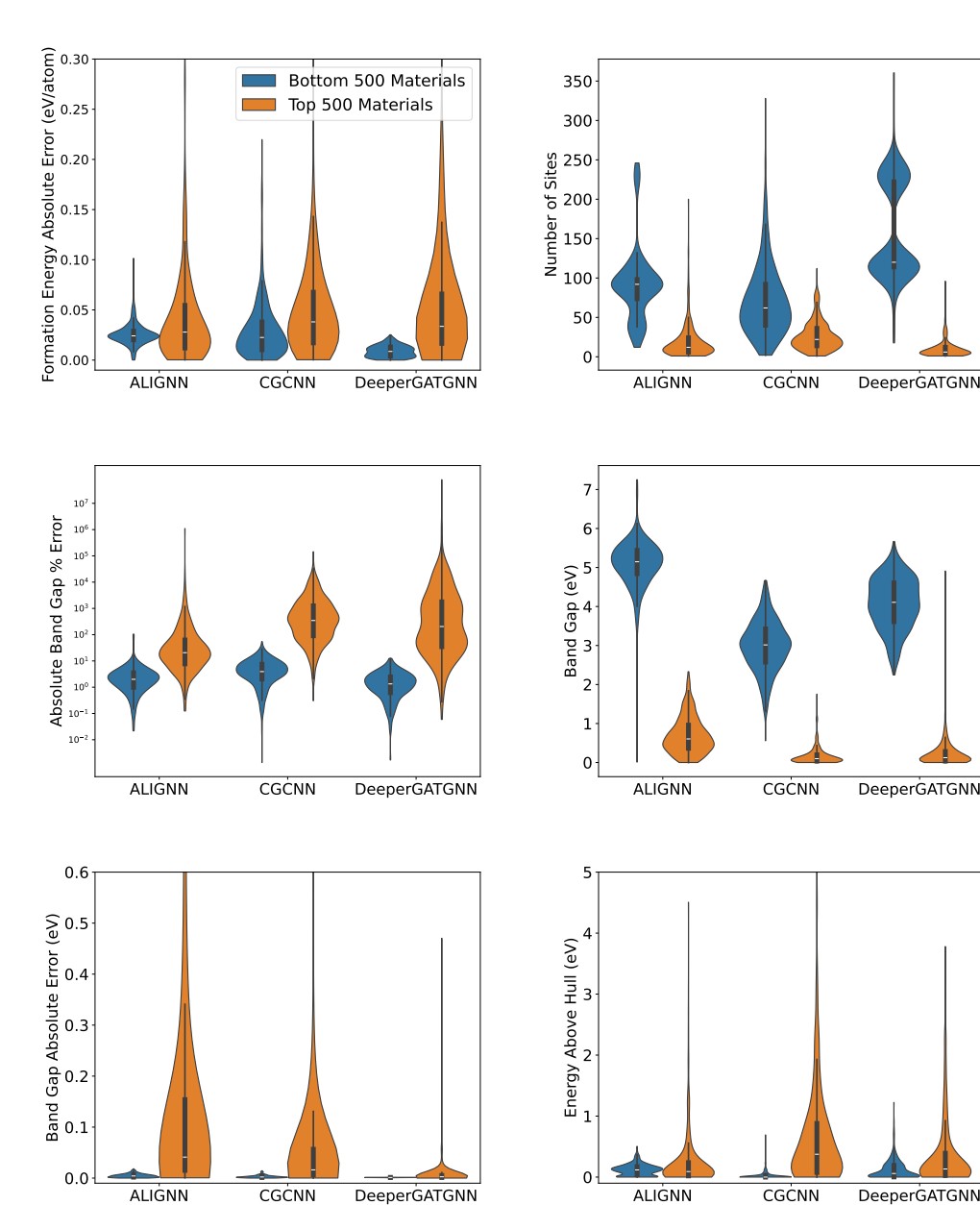

Figure 12: Property prediction worsens on rare materials across models. Top: Analysis of formation energy models, showing larger predictive errors on the 500 most rare materials compared to the bottom 500 materials as computed by the Vendiscope. (Left). The rare materials correspond to those with less sites in the unit cell (Right). Middle: Analysis of band-gap prediction models on non-conducting materials. Predictive errors are higher for rare materials compared to common materials, as shown by violin plots of error distribution (Left). Y-axis is logarithmic for display. The rare materials correspond to those with smaller band-gaps (Right). Bottom: Analysis of band-gap prediction models on conductors. Prediction errors are significantly higher for rare materials than common ones (Left). For all models except ALIGNN, rare materials correspond to those with higher energies above the hull (Right). All distributions are statistically distinct as measured by Mann-Whitney U Test ($p < 0.01$) unless otherwise specified.

them to run the Vendiscope. In Figure 12, we show that the error associated with formation energy prediction is significantly higher for rare materials. The rare materials, as shown in 12, tend to have a smaller number of sites in their unit cell.

To characterize model behavior further, we partitioned materials into conductors (band gap $= 0$ eV) and non-conductors (band gap $\neq 0$). Applying the Vendiscope to the embeddings from each group separately shows that model performance worsens significantly on rare materials. Across all models, rare materials are shown to have distinct physical properties from their bottom-scoring counterparts. One-sided Mann-Whitney U tests confirm that rare non-conductors have lower band gaps than common materials. The tests also confirm that rare conductors have large energies above the hull for both the CGCNN and DeeperGATGNN models.

The failure of models to generalize to rare materials is unsurprising - previous work by Li et al. (2023) also observe strong performance on redundant materials and poorer performance elsewhere. Our findings motivate future data collection in the Materials Project database. Researchers should aim to add smaller materials, semi-conductors, and less stable conductors to improve model performance.

**The Vendiscope detects duplicate crystals in the Materials Project database.** We also apply the Vendiscope to detect near-duplicates in the Materials Project database in the two embedding spaces from ALIGNN. The first embedding space is the one implied by formation energy prediction.

Using Algorithm 2, we identify that $148,907$ materials (87.9% of the dataset) are near-duplicates at a similarity threshold of $s = 0.9$, decreasing only to $121,683$ at a stricter threshold of $s = 0.95$. The second embedding space is the one corresponding to band gap prediction. Among conductors in this space, $67,910$ materials are near-duplicates at $s = 0.9$, with $52,684$ remaining near-duplicates at $s = 0.95$. For non-conductors, $78,643$ materials are near-duplicates at $s = 0.9$, and $65,891$ materials remain above the stricter threshold of $0.95$.

With the Vendiscope, we are able to find all of these near-duplicates rapidly: in all embedding spaces, we only need to compute 19% of all pair-wise similarities in the Materials Project database. Alternative approaches to identifying materials with similar structures rely on computing all pair-wise similarities and require manual inputs. For example, the Materials Project database compares carefully curated coordination site fingerprints across all materials to identify crystals with similar atomic arrangements and bonding patterns.

### A.4 IMAGE GENERATIVE MODEL ANALYSIS

#### A.4.1 EXPERIMENTAL SETTINGS

We employ image embeddings from the DINOv2 ViT-L/14 network (Oquab et al., 2023). We choose the DINOv2 network based on the findings from Stein et al. (2023) that showed it provides the best evaluations of generative models. In all analyses, we use a cosine similarity kernel and a Vendi Score order of $q = 0.1$. Duplicates are identified with a search range of $m = 10,000$ and a similarity threshold of $s = 0.9$, which corresponds to computing only 33% of all pairwise similarities on CIFAR-10. To analyze the generative models from Stein et al. (2023), we run the Vendiscope on the DINOv2 embeddings of $50,000$ generated images from each model.

#### A.4.2 ADDITIONAL ANALYSIS OF IMAGE GENERATIVE MODELS

Specific examples of the varying degrees of memorization for rare and common samples from the iDDPM-DDIM model are displayed in Fig. 14 (Nichol & Dhariwal, 2021). The rare samples in the training dataset are not represented in the generated dataset, whereas the model generates almost exact replicas of common samples.

We also find that models whose pattern of memorization can be explained by the Vendiscope's ranking of training data create the highest-quality images (Figure 15). These models memorize common images and do not recreate the rare training samples in their outputs. Models that do not follow this pattern of memorization, such as the LOGAN model, do so at the cost of creating high-quality images. Finally, in Figure 16, we run the Vendiscope on the generated outputs from each model. Across models, the generated samples that receive the lowest Vendiscope scores are those

| Method | Input | Complexity | |
|--------|-------|-----------|------|
| | | **Time** | **Space** |
| MMSeqs2 | Protein Sequences | $O(N)$ | $O(NL)$ |
| knnProtT5 | Protein Embeddings | $O(N\log N)$ | $O(ND)$ |
| MinHash | Raw Text | $O(KT^2N)$ | $O(NK)$ |
| RETSim | Text Embeddings | $O(ND)$ | $O(ND)$ |
| The Vendiscope | Any Embedding | $O(Nm+ND^2)$ | $O(ND)$ |

Table 1: A comparison of various de-duplication methods for a dataset with $N$ samples. For protein sequence databases, we denote $L$ as the maximum protein sequence length. For embedding-based methods, we denote $D$ as the dimensionality of each sample's embedding. For MinHash, we denote $K$ as the number of hashing functions used, and $T$ as the maximal number of tokens in a document. In the Vendiscope, we denote $m$ as the search-range used in Algorithm 2.

that lie closest to the training data. This suggests we can use the Vendiscope to identify model memorization, even in the absence of training data.

Our findings span popular model architectures, including diffusion models, GANs, VAEs, and flows. In all, we tested 8 GAN models: ACGAN (Odena et al., 2017), BigGAN (Brock et al., 2019), LOGAN (Wu et al., 2019), ReACGAN (Kang et al., 2021), MHGAN, (Turner et al., 2019), WGAN-GP (Gulrajani et al., 2017), StyleGAN2-ada (Karras et al., 2020), and StyleGAN2-XL (Sauer et al., 2022). Additional models tested include NVAE (Vahdat & Kautz, 2020), RESFLOW (Chen et al., 2019), and the three diffusion models iDDPM-DDIM (Nichol & Dhariwal, 2021) PFGM++ (Xu et al., 2023), and LSGM-ODE (Vahdat et al., 2021).

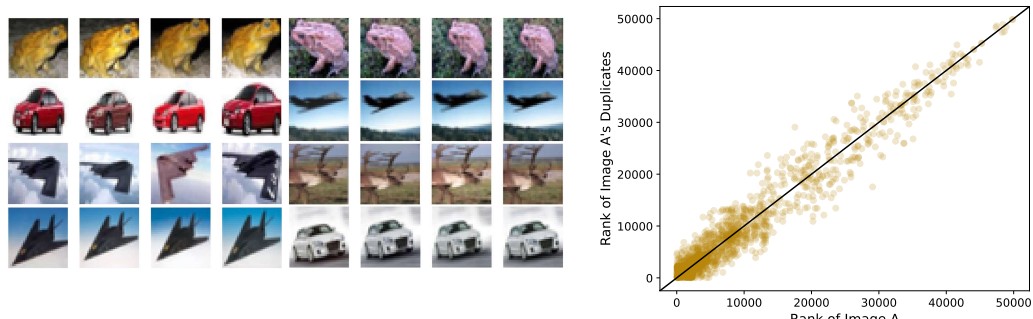

Figure 13: **The Vendiscope Helps Detect Near-Duplicates** Left: Selected near-duplicates present in the training CIFAR10 dataset. Right: The Vendiscope ranks of each pair of near-duplicates are concentrated along the diagonal, demonstrating that similar images contribute similarly to a dataset's overall diversity. A total of $955$ images are near-duplicates.

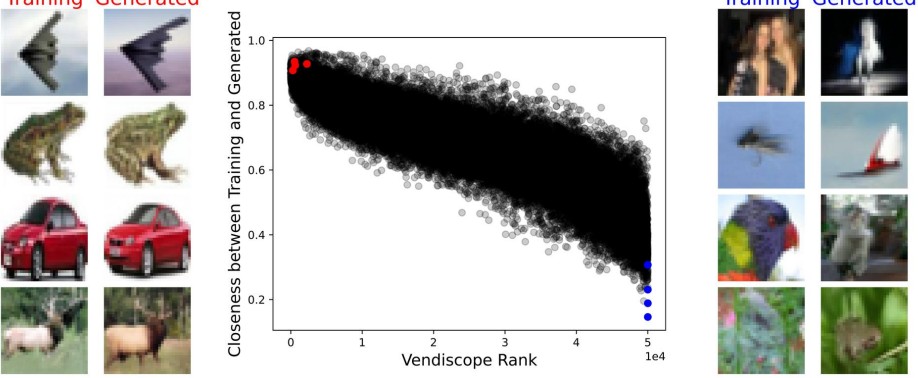

Figure 14: CIFAR-10 training data Vendiscope scores are strongly correlated with their degree of memorization. Results shown for iDDPM-DDIM model. Left: Redundant training samples, those with low contributions to diversity, are memorized by the generative model. Samples are marked in red on the center plot. Right: Rare samples, those with high diversity contributions, are not memorized. Samples are marked in blue on the center plot.

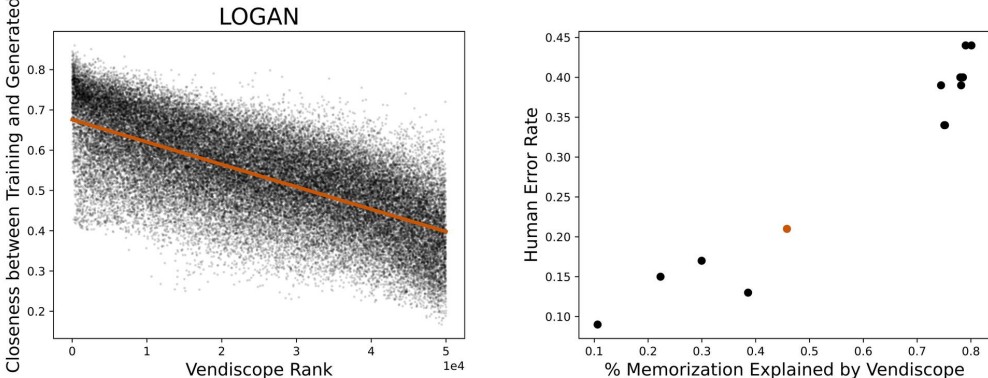

Figure 15: Image fidelity is linked to memorization of common samples. Left: Scatter plot showing the ranking of samples by Vendiscope weights for CIFAR-10 training data against their degree of memorization by the LOGAN model. Line of best fit shows correlation between the two. Right: Scatter plot of the Human Error Rate for all 13 models (LOGAN highlighted in orange) against what % of the Memorization can be explained by the Vendiscope's ranking of the training data. % Memorization explained is measured by computing the $R^2$ between the Vendiscope's ranking of CIFAR-10 training data and the closeness to the nearest generated sample.

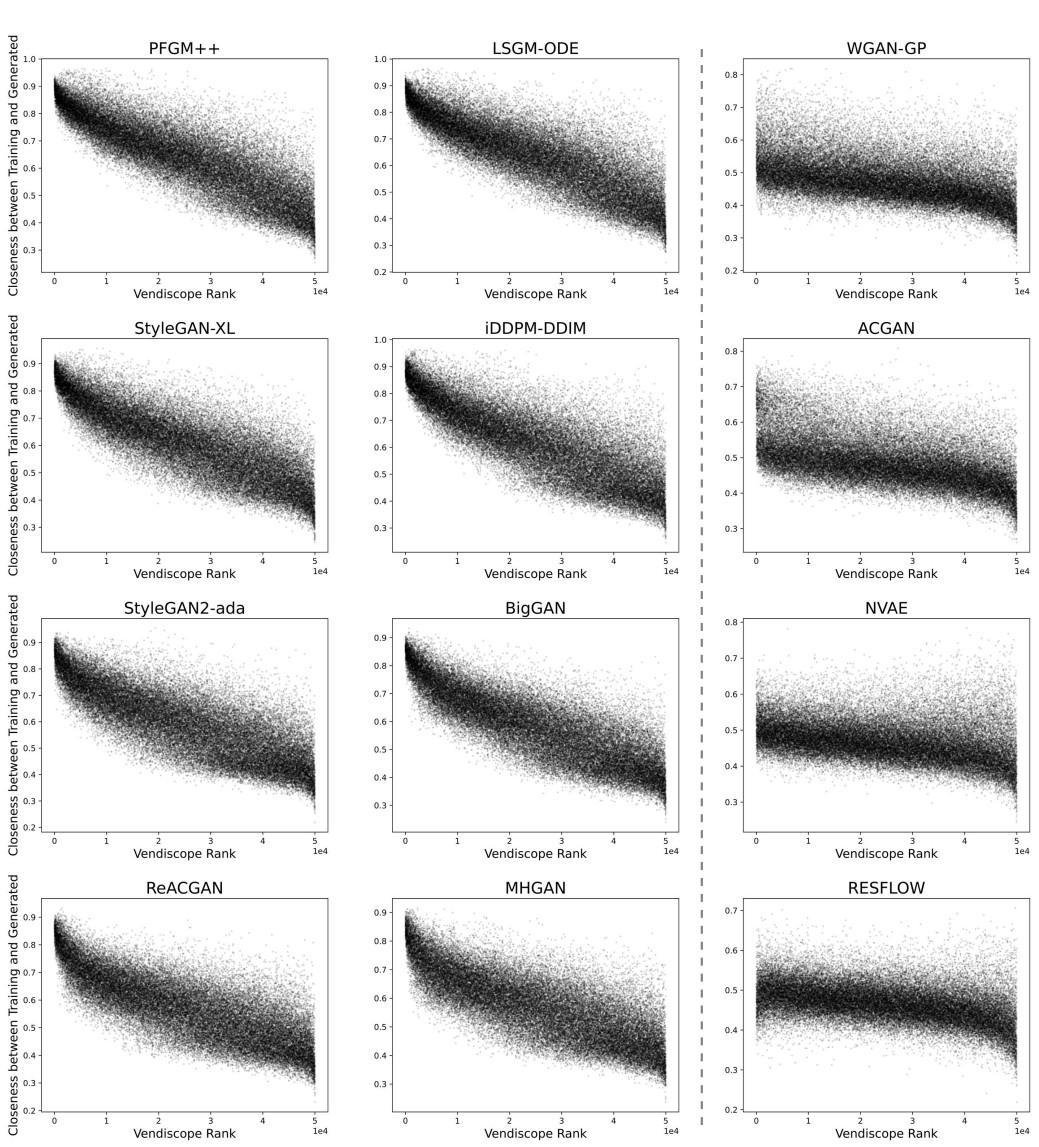

Figure 16: Memorization is also correlated with the Vendiscope rank of CIFAR-10 synthetic images for all tested various image generative models. Memorization of a generated image is measured as its highest similarity to any sample in the training set. Models for which the correlation is weaker are in the third column.

