# OpenReview forum: "The Vendiscope: An Algorithmic Microscope For Data Collections"
_ICLR.cc/2026/Conference — Submitted to ICLR 2026_

### Official Review · Reviewer_vUBK · 2025-10-25

**Soundness:** 2
**Presentation:** 2
**Contribution:** 3
**Rating:** 6
**Confidence:** 2

**Summary:**

This paper introduces the concept of algorithmic microscopes, tools designed to reveal hidden structure in both dataset composition and model behavior. An algorithmic micro.scope emphasizes understanding - helping researchers understand the contents of their data andwhere their models fail.

**Strengths:**

1.They demonstrate the Vendiscope's contribution scores can help identify outliers and near-duplicates in linear time.
2.They show how the same framework can be used to evaluate machine learning models -both predictive and generative.
3.They apply the Vendiscope to the 250 million sequences composing the protein universe.where it identifies >80% redundant data points at a 90% similarity threshold.

**Weaknesses:**

1. Adding a figure to show the overview of the framework will help readers to quickly understand your contribution.
    2. The Vendiscope's outputs are fundamentally dependent on the choice of the similarity kernel and the data embeddings. The claim that the method works "in any domain where similarity can be defined" is true in principle, but the practical utility and interpretability of the results are highly contingent on the quality and semantics of the chosen similarity function. An analysis of this sensitivity is crucial.
    3. The selected generative model architectures are not the recent published works. As mentioned in  A4.2, all of them are published before 2021. I believe the author should compare more methods in the recent two years.
    4. More explanation about figures is needed. For example, how to understand the statements that “the most common images generated by each model are those that are increasingly similar to the training data.” shown in Figure 13?

**Questions:**

See weakness above.

---

> ### Author Response · Authors · 2025-11-21
> **Response to Reviewer vUBK**
>
> We appreciate your thoughtful feedback and comments.
>
> > Adding a figure to show the overview of the framework will help readers to quickly understand your contribution.
>
> Thank you for your advice! We are considering options for such a schematic.
>
> > The Vendiscope's outputs are fundamentally dependent on the choice of the similarity kernel and the data embeddings. The claim that the method works "in any domain where similarity can be defined" is true in principle, but the practical utility and interpretability of the results are highly contingent on the quality and semantics of the chosen similarity function. An analysis of this sensitivity is crucial.
>
> We agree that the Vendiscope is dependent on the choice of similarity kernel and features used. Different similarity functions, by design, result in different notions of rarity. We show this in the added Figures 7 and 8. The Vendiscope’s rankings should change if we use features and kernels that prioritize different aspects of the data collection. Therefore, robustness to the choice of embedding and kernel function is not a desired feature.
>
> Across all of our experiments, we make sure to use embedding models that are known to capture the semantics of the given domain. For example, the DinoV2 model was determined in Stein et al, 2023 to be the best embedding model at aligning with human perception. We therefore used DinoV2 in all of our image experiments.
>
> > The selected generative model architectures are not the recent published works. As mentioned in A4.2, all of them are published before 2021. I believe the author should compare more methods in the recent two years.
>
> Indeed, most of the selected models are over three years old. However, our experiment aims to show that memorization is a systemic limitation of generative models rather than an artifact of a specific architecture. We compare against 13 models spanning multiple classes, which we believe is sufficient to demonstrate our findings.
>
> > More explanation about figures is needed. For example, how to understand the statements that “the most common images generated by each model are those that are increasingly similar to the training data.” shown in Figure 13?
>
> Thank you for highlighting this - our updated draft contains more explanation for figures like Figure 13.
>
> Citation:
>
> Stein, G., Cresswell, J., Hosseinzadeh, R., Sui, Y., Ross, B., Villecroze, V., ... & Loaiza-Ganem, G. (2023). Exposing flaws of generative model evaluation metrics and their unfair treatment of diffusion models. Advances in Neural Information Processing Systems, 36, 3732-3784.

---

> > ### Author Response · Authors · 2025-12-01
> > **Follow-up Response to Reviewer vUBK**
> >
> > Dear Reviewer vUBK,
> >
> > We are following up to ask whether you have any further questions or comments. We believe we have addressed all your feedback and would request that you update your score, unless you have further questions/comments you would like us to address.
> >
> > Thank you,
> >
> > Authors

---

### Official Review · Reviewer_TgHy · 2025-10-31

**Soundness:** 3
**Presentation:** 4
**Contribution:** 3
**Rating:** 8
**Confidence:** 4

**Summary:**

This paper proposes a method for performing analysis of datasets to uncover factors such as data samples that contribute least to diversity.  Specifically, this paper leverages Vendi scores, a metric of diversity based on a given similarity metric for a data set.  The proposed methodology is evaluated on several data sets.  Results on protein data show that the method can identify rare sequences, for which models such as AlphaFold perform most poorly.  Redundant protein sequences can be clustered, showing strong performance relative to the domain baseline of MMseqs2.  In the domain of image generative models, the proposed method can identify near-duplicates in CIFAR-10 and in generated images, and with respect to generated images, models that do not generate near-duplicates produce lower quality output as measured by human judgment.  As well, near-duplicates can be found in cross-matches between CIFAR-10 and generated images, indicating memorization, and the paper demonstrates a strong negative correlation between training examples with low score (contributing least to data set diversity) and likelihood of being reproduced/memorized by the models.

**Strengths:**

Demonstrates applicability over three different domains (one in appendix).  Findings have relevance/applicability to each domain, eg strong clustering results on protein sequences relative to MMseqs2, and proper evaluation metrics in the presences of duplicates for image generative modeling.

**Weaknesses:**

Would be interested to see (possibly deferred to the appendix) slightly more discussion on any potential issues in generating the Vendi scores.  For instance, how was it decided whether to use q=0.1 or 0.5, and how sensitive were the results to the value of q.  Or, how sensitive are the results to the particular chosen data embedding?

**Questions:**

see weaknesses above

---

> ### Author Response · Authors · 2025-11-21
> **Response to Reviewer TgHy**
>
> Thank you for your feedback!
>
> > For instance, how was it decided whether to use q=0.1 or 0.5, and how sensitive were the results to the value of q. Or, how sensitive are the results to the particular chosen data embedding?
>
> We have provided an updated draft to include Figures 6,7, and 8 which justify our choices of q as well as the Vendiscope’s sensitivity to the choice of similarity function.
>
> We have also added additional discussion in our paper about the choice of embedding model and choice of q in section 2.3. Please look at our overall response for additional details!

---

> > ### Author Response · Authors · 2025-12-01
> > **Follow-up Response to Reviewer TgHy**
> >
> > Dear Reviewer TgHy,
> >
> > We are following up to ask whether you have any further questions or comments. We believe we have addressed all your feedback and would request that you update your score, unless you have further questions/comments you would like us to address.
> >
> > Thank you,
> >
> > Authors

---

### Official Review · Reviewer_YqP7 · 2025-11-01

**Soundness:** 2
**Presentation:** 2
**Contribution:** 1
**Rating:** 2
**Confidence:** 3

**Summary:**

The paper introduces the concept of algorithmic microscope, termed vendiscope, that uses probability-weighted vendi scores to measure data-point contribution to the diversity in the dataset. Using this framework, the paper investigates the diversity of protein sequences and Alphafold’s inability to predict rare proteins, and also highlights on the generative model’s memorization aspect on CIFAR-10 data.

**Strengths:**

+ The paper is overall easy to understand.

+ The paper conducted a rich amount of experiments.

**Weaknesses:**

- The main weakness lies in the contribution of the paper. Using vendi scoring to quantify the diversity of datasets does not seem like a significant contribution. The paper neither leads to any new finding using this scheme. It is well known that Alphafold struggles to predict structure for proteins with low homologs. The generative model’s memorization aspect is also well-known.

- It is not clear why CIFAR-10 is used for images, it is a small image dataset with limited diversity. The images are of very small resolution and there is no scene-centric images. I think ImageNet would have been the best choice.

- Given the choice of small dataset for images, I am not sure which UniProt dataset it used for Proteins. For comparison with protein sequence clustering, only MMSeq2 was used despite the existence of many other widely-used methods like CD-Hit or deep embedding based methods.

- The paper claims that it performed experiments on proteins, images, and materials (L478) where it clearly did not do anything on materials. These makes the work seem incomplete.

- The organization of the paper is poor. The related work, which is supposed to serve as the background for the work, should come before method.

**Questions:**

1. Why CIFAR-10 for images instead of ImageNet?
2. Which UniProt dataset was used?
3. How does the protein sequence clustering application work compared to the deep embedding clustering approaches?

---

> ### Author Response · Authors · 2025-11-21
> **Response to Reviewer YqP7 (Part 1)**
>
> Thank you for your comments!
>
> > The main weakness lies in the contribution of the paper. Using vendi scoring to quantify the diversity of datasets does not seem like a significant contribution. The paper neither leads to any new finding using this scheme. It is well known that Alphafold struggles to predict structure for proteins with low homologs. The generative model’s memorization aspect is also well-known.
>
> We would like to clarify that the Vendiscope is not a diversity metric and does not attempt to measure the diversity of a dataset. The Vendi Score, introduced in Friedman and Dieng (2023), is a global summary statistic that measures the overall diversity of a dataset. In contrast, the Vendiscope quantifies the per-datapoint contribution to this global statistic, allowing us to examine dataset composition and model behavior.
>
> In our various experiments across domains, we show how the Vendiscope can reveal insights into the contents of datasets and model failure modes. Such analyses would otherwise require ad-hoc, domain-specific methods. For example, while it is known that Alphafold struggles on proteins with low homologs, the Vendiscope provides a scalable method to identify these proteins without requiring separate evolutionary profiles or alignment scores. On images, it is known that generative models can memorize. However, the Vendiscope reveals the types of images that are memorized and provides strong evidence for how sample rarity is linked to memorization. Previous work has often focused only on duplicates and their role in memorization (e.g. Lee et al., 2021).
>
>
> > It is not clear why CIFAR-10 is used for images, it is a small image dataset with limited diversity. The images are of very small resolution and there is no scene-centric images. I think ImageNet would have been the best choice.
>
> Although CIFAR-10 does have limitations, CIFAR-10 remains a standard benchmark in the literature for studying overfitting and memorization in generative models. Its size allows us to perform a large number of detailed experiments across many generative models and detect their memorization profiles.
>
> We have updated the draft to include experiments of how the Vendiscope on ImageNet classes can reveal outliers/mislabelled images (Figures 7 and 8)
>
> > Given the choice of small dataset for images, I am not sure which UniProt dataset it used for Proteins. For comparison with protein sequence clustering, only MMSeq2 was used despite the existence of many other widely-used methods like CD-Hit or deep embedding based methods.
>
> We mention the exact version of the UniProt dataset in Appendix section A.2.1. This represents the entire database, containing over 250 million protein sequences.
>
> We also agree that many clustering methods exist for protein sequences, including CD-HIT and other embedding clustering methods. The goal of these experiments was not to benchmark clustering algorithms, but rather to evaluate whether the Vendiscope can recover biologically meaningful protein sequence clusters under our domain-agnostic framework. We benchmark against MMSeqs2 because it is among the fastest and most popular tools today. The UniProt team uses MMSeqs2 to curate smaller versions of the dataset like UniRef90 and UniRef50. MMSeqs2 is also shown to achieve a 60x speed-up over CD-Hit (Steinegger & Soding, 2018). Our experiments with MMSeqs2 provide a strong representative baseline for evaluating the Vendiscope’s ability.
>
> New embedding clustering methods are also being developed - we had therefore provided a comparison of the approaches taken by the Vendiscope and one popular framework (knnProtT5) in Section 4 and in Table 1.

---

> > ### Author Response · Authors · 2025-11-21
> > **Response to Reviewer YqP7 (Part 2)**
> >
> > > The paper claims that it performed experiments on proteins, images, and materials (L478) where it clearly did not do anything on materials. These makes the work seem incomplete.
> >
> > The experiments on the materials science dataset are in Appendix Section A.3. We point to this section in the first paragraph of Section 3.
> >
> > > The organization of the paper is poor. The related work, which is supposed to serve as the background for the work, should come before method.
> >
> > We appreciate the comment. We chose to present the method before the related work because the Vendiscope introduces a framework that does not fall cleanly into any single prior category (e.g. outlier detection, de-duplication, or memorization detection). By defining the method first and showing our experiments, we hope the reader is able to have context for how the Vendiscope connects to each of these areas before we outline related works. We are happy to discuss further if you believe that it is difficult to understand our contributions with this organization.
> >
> > Citations:
> >
> > Friedman, D. and Dieng, A. B. (2023). The Vendi Score: A Diversity Evaluation Metric for Machine Learning. Transactions on Machine Learning Research.
> >
> > Lee, K., Ippolito, D., Nystrom, A., Zhang, C., Eck, D., Callison-Burch, C., & Carlini, N. (2022, May). Deduplicating training data makes language models better. In Proceedings of the 60th Annual Meeting of the Association for Computational Linguistics.
> >
> > Steinegger, M., & Söding, J. (2018). Clustering huge protein sequence sets in linear time. Nature communications, 9(1), 2542.

---

> ### Author Response · Authors · 2025-12-01
> **Follow-up Response to Reviewer YqP7**
>
> Dear Reviewer YqP7,
>
> We are following up to ask whether you have any further questions or comments. We believe we have addressed all your feedback and would request that you update your score, unless you have further questions/comments you would like us to address.
>
> Thank you,
>
> Authors

---

### Official Review · Reviewer_F2E2 · 2025-11-03

**Soundness:** 2
**Presentation:** 2
**Contribution:** 3
**Rating:** 4
**Confidence:** 3

**Summary:**

This paper presents Vendiscope, an algorithmic microscope that quantifies each datapoint’s contribution to dataset diversity using probability-weighted Vendi Scores. It efficiently detects duplicates, rare samples, and memorization patterns across large datasets like UniProt and CIFAR-10. The approach offers a scalable, domain-agnostic tool for understanding data composition and model performance beyond conventional accuracy metrics.

**Strengths:**

(1) This paper introduces a unified, scalable framework to quantify each datapoint’s contribution to dataset diversity across multiple domains.

(2) It effectively bridges data analysis and model diagnosis by revealing redundancy, rarity, and memorization patterns using a single interpretable metric.

**Weaknesses:**

See questions

**Questions:**

(1) Is there any justification or evidence of how the kernel type or hyperparameters affect the interpretability and stability of Vendiscope scores, as a positive semi-definite similarity kernel is assumed?

(2) It is not quantified how this approximation impacts accuracy or reproducibility of the Vendiscope rankings.

---

> ### Author Response · Authors · 2025-11-21
> **Response to Reviewer F2E2**
>
> Thank you for your feedback!
>
> > Is there any justification or evidence of how the kernel type or hyperparameters affect the interpretability and stability of Vendiscope scores, as a positive semi-definite similarity kernel is assumed?
>
> We have updated our draft to include experiments (Figures 6, 7, and 8) showing how the choice of kernel and choice of order $q$ can affect the rankings produced by the Vendiscope. Please look at our overall response for additional details!
>
> > It is not quantified how this approximation impacts accuracy or reproducibility of the Vendiscope rankings.
>
> Can you please clarify the approximation you are referring to? We would be more than happy to provide additional justification for the accuracy of the Vendiscope once this is made clear.

---

> > ### Author Response · Authors · 2025-12-01
> > **Follow-up Response to Reviewer F2E2**
> >
> > Dear Reviewer F2E2,
> >
> > We are following up to ask whether you have any further questions or comments. We believe we have addressed all your feedback and would request that you update your score, unless you have further questions/comments you would like us to address.
> >
> > Thank you,
> >
> > Authors

---

### Author Response · Authors · 2025-11-21
**Response to Reviewers**

Thank you to the reviewers for their thoughtful comments. The reviewers made a number of great suggestions. We have incorporated these into our draft and uploaded an updated version of the paper.

In particular, reviewers F2E2, TgHy, and vUBK all noted that the Vendiscope can be sensitive to the choice of kernel function, embedding model, and Vendi Score order $q$. We have therefore added 3 additional figures.

Figure 6: A synthetic 2D experiment highlighting how the Vendiscope measures rarity for different choices of $q$. We show that for all finite values of $q$, the Vendiscope measures rarity similarly, with small values of $q$ being the most sensitive to individual samples. For $q=\infty$, the Vendiscope focuses on the distance of samples from the largest mode. We have updated Section 2.3 to include this analysis.

Figures 7 and 8: Demonstration of how the choice of embedding and kernel function will lead to different Vendiscope rankings on ImageNet classes. Embeddings generated from DINOv2 and Inception focus on semantics, whereas alternative methods focus on colors.

We have also updated the explanation for certain figures in the text. All updates are highlighted in red.

We look forward to continuing to engage with the reviewers throughout the rebuttal period!

---

### Meta-Review · Area_Chair_b5Dj · 2026-01-07

**Summary:**

Reviewers agree the idea is novel and potentially broadly useful, with a strong positive review emphasizing convincing cross-domain demonstrations (proteins, images, materials). The main concerns raised across reviews are:
- sensitivity to the choice of embedding and similarity kernel and how practitioners should choose these;
- hyperparameter choices (notably the Vendi order q, and kernel / kNN settings) and stability of rankings;
- clarity around approximation and reproducibility of the ranking;
- presentation and experimental choices (one review also contained multiple factual misunderstandings, limiting its usefulness).

**Reviewer Concerns:**

### Addressed by rebuttal / discussion

- Kernel, embedding, and q sensitivity: Authors clarified that different embeddings and similarity kernels intentionally produce different “rarity” notions, and added/pointed to sensitivity analyses and updated figures/discussion to make this explicit.

- Clarity on experimental coverage: Concerns about missing domains appear addressed by pointing reviewers to the relevant sections/appendix and clarifying what experiments were run.

- Presentation: Several clarifications were provided (more explanations around key figures and methodological choices).

### Still outstanding

- Actionable guidance for users: Even if dependence on embedding/kernel is expected, the paper would benefit from clearer best practices, diagnostics, or default recommendations for selecting embeddings/kernels and q in common regimes.

- Approximation and reproducibility: The discussion would be stronger with a more explicit quantification of how approximation impacts ranking stability (for example, rank correlation vs compute, and when failures occur).

- Empirical baselines breadth: Some requests (for additional comparisons/broader experimental settings) are reasonable but not all are necessary for acceptance if the paper clearly scopes what Vendiscope is and is not.

**Reviewer Scores:**

Reviewer TgHy: Likely unchanged (stays 8). The reviewer’s remaining questions are mainly about guidance and sensitivity, which the rebuttal partially addressed.

Reviewer vUBK: Likely increases modestly to 7 with the added clarifications and sensitivity discussion, since the core contribution was viewed positively but previously borderline.

Reviewer F2E2: Could stay at 4 or move to 5 if they accept the added kernel/q analyses and clarifications, but may remain cautious pending clearer quantification of approximation effects.

Reviewer YqP7 (2, moderate confidence): Likely unchanged because the review contains several factual inaccuracies / missed content, and does not engage deeply with the author after clarification.

---

### Decision · Program_Chairs · 2026-01-26

Reject